# Making Pairwise Binary Graphical Models Attractive

**Nicholas Ruozzi**
Institute for Data Sciences and Engineering
Columbia University
New York, NY 10027
nr2493@columbia.edu

**Tony Jebara**
Department of Computer Science
Columbia University
New York, NY 10027
jebara@cs.columbia.edu

## Abstract

Computing the partition function (i.e., the normalizing constant) of a given pairwise binary graphical model is NP-hard in general. As a result, the partition function is typically estimated by approximate inference algorithms such as belief propagation (BP) and tree-reweighted belief propagation (TRBP). The former provides reasonable estimates in practice but has convergence issues. The later has better convergence properties but typically provides poorer estimates. In this work, we propose a novel scheme that has better convergence properties than BP and provably provides better partition function estimates in many instances than TRBP. In particular, given an arbitrary pairwise binary graphical model, we construct a specific "attractive" 2-cover. We explore the properties of this special cover and show that it can be used to construct an algorithm with the desired properties.

## 1   Introduction

Graphical models provide a mechanism for expressing the relationships among a collection of variables. Many applications in computer vision, coding theory, and machine learning can be reduced to performing statistical inference, either computing the partition function or the most likely configuration, of specific graphical models. In general models, both of these problems are NP-hard. As a result, much effort has been invested in designing algorithms that can approximate, or in some special cases exactly solve, these inference problems.

The belief propagation algorithm (BP) is an efficient message-passing algorithm that is often used to approximate the partition function of a given graphical model. However, BP does not always converge, and so-called convergent message-passing algorithms such as tree reweighted belief propagation (TRBP) have been proposed as alternatives to BP. Such convergent message passing algorithms can be viewed as dual coordinate-descent schemes on a particular convex upper bound on the partition function [1]. While TRBP-style message-passing algorithms guarantee convergence under suitable message-passing schedules, finding the optimal message-passing schedule can be cumbersome or impractical depending on the application, and TRBP often performs worse than BP in terms of estimating the partition function.

The primary goal of this work is to study alternatives to BP and TRBP that have better convergence properties than BP and approximate the partition function better than TRBP. To that end, the so-called "attractive" graphical models (i.e., those models that do not contain frustrated cycles) stand out as a special case. Attractive graphical models have desirable computational properties: Weller and Jebara [2, 3] describe a polynomial time approximation scheme to minimize the Bethe free energy of attractive models (note that BP only guarantees convergence to a local optimum). In addition, BP has much better convergence properties on attractive models than on general pairwise binary models [4, 5].

In this work, we show how to approximate the inference problem over a general pairwise binary graphical model as an inference problem over an attractive graphical model. Similar in spirit to the work of Bayati et al. [6] and Ruozzi and Tatikonda [7], we will use graph covers in order to better understand the behavior of the Bethe approximation with respect to the partition function. In particular, we will show that if a graphical model is strictly positive and contains even one frustrated cycle, then there exists a choice of external field and a 2-cover without frustrated cycles whose partition function provides a strict upper bound on the partition function of the original model. We then show that the computation of the Bethe partition function can approximated, or in some cases found exactly, by computing the Bethe partition function over this special cover. The required computations are easier on this "attractive" graph cover as computing the MAP assignment can be done in polynomial time and there exists a polynomial time approximation scheme for computing the Bethe partition function.

We illustrate the theory through a series of experiments on small models, grid graphs, and vertex induced subgraphs of the Epinions social network[1], . All of these models have frustrated cycles which make the computation of their partition functions, marginals, and most-likely configurations exceedingly difficult. In these experiments, the proposed scheme converges significantly more frequently than BP and provides a better estimate of the partition function than TRBP.

## 2    Prerequisites

We begin by reviewing pairwise binary graphical models, graph covers, the Bethe and TRBP approximations, and recent work on lower bounds.

### 2.1    Pairwise Binary Graphical Models

Let $f : \{0,1\}^n \to \mathbb{R}_{\geq 0}$ be a non-negative function. A function $f$ factors with respect to a graph $G = (V, E)$, if there exist potential functions $\phi_i : \{0,1\} \to \mathbb{R}_{\geq 0}$ for each $i \in V$ and $\psi_{ij} : \{0,1\}^2 \to \mathbb{R}_{\geq 0}$ for each $(i,j) \in E$ such that

$$f(x_1, \ldots, x_n) = \prod_{i \in V} \phi_i(x_i) \prod_{(i,j) \in E} \psi_{ij}(x_i, x_j).$$

The graph $G$ together with the collection of potential functions $\phi$ and $\psi$ define a graphical model that we will denote as $(G; \phi, \psi)$. For clarity, we will often denote the corresponding function as $f^{(G;\phi,\psi)}(x)$. For a given graphical model $(G; \phi, \psi)$, we are interested in computing the partition function $Z(G; \phi, \psi) \triangleq \sum_{x \in \{0,1\}^{|V|}} \prod_{i \in V} \phi_i(x_i) \prod_{(i,j) \in E} \psi_{ij}(x_i, x_j)$.

We will also be interested in computing the maximum value of $f$, sometimes referred to as the MAP problem. The problem of computing the MAP solution can be converted into the problem of computing the partition function by adding a temperature parameter, $T$, and taking the limit as $T \to 0$.

$$\max_x f^{(G;\phi,\psi)}(x) = \lim_{T \to 0} Z(G; \phi^{1/T}, \psi^{1/T})^T$$

Here, $\phi^{1/T}$ is the collection of potentials generated by taking each potential $\phi_i(x_i)$ and raising it to the $1/T$ power for all $i \in V, x_i \in \{0,1\}$.

### 2.2    Graph Covers

Graph covers have played an important role in our understanding of statistical inference in graphical models [8, 9]. Roughly speaking, if a graph $H$ covers a graph $G$, then $H$ looks locally the same as $G$.

**Definition 2.1.** *A graph $H$ **covers** a graph $G = (V, E)$ if there exists a graph homomorphism $h : H \to G$ such that for all vertices $i \in G$ and all $j \in h^{-1}(i)$, $h$ maps the neighborhood $\partial j$ of $j$ in $H$ bijectively to the neighborhood $\partial i$ of $i$ in $G$.*

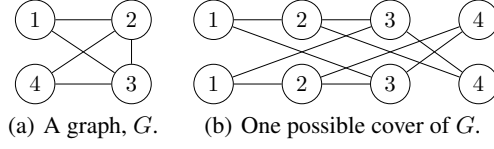

(a) A graph, $G$.      (b) One possible cover of $G$.

Figure 1: An example of a graph cover. The nodes in the cover are labeled for the node that they copy in the base graph.

If $h(j) = i$, then we say that $j \in H$ is a copy of $i \in G$. Further, $H$ is said to be an $M$-cover of $G$ if every vertex of $G$ has exactly $M$ copies in $H$. For an example of a graph cover, see Figure 1. For a connected graph $G = (V, E)$, each $M$-cover consists of $M$ copies of each of the variable nodes of $G$ with an edge joining each distinct copy of $i \in V$ to a distinct copy of $j \in V$ if and only if $(i, j) \in E$.

To any $M$-cover $H = (V^H, E^H)$ of $G$ given by the homomorphism $h$, we can associate a collection of potentials: the potential at node $i \in V^H$ is equal to $\phi_{h(i)}$, the potential at node $h(i) \in G$, and for each $(i, j) \in E^H$, we associate the potential $\psi_{h(i,j)}$. In this way, we can construct a function $f^{(H;\phi^H, \psi^H)} : \{0, 1\}^{M|V|} \to \mathbb{R}_{\geq 0}$ such that $f^{(H;\phi^H, \psi^H)}$ factorizes over $H$. We will say that the graphical model $(H; \phi^H, \psi^H)$ is an $M$-cover of the graphical model $(G; \phi, \psi)$ whenever $H$ is an $M$-cover of $G$ and $\phi^H$ and $\psi^H$ are derived from $\phi$ and $\psi$ as above.

## 2.3 The Bethe Partition Function

The Bethe free energy is a standard approximation to the so-called Gibbs free energy that is motivated by ideas from statistical physics. TRBP and more general reweighted belief propagation algorithms take advantage of a similar approximation.

For $\tau$ in the *local marginal polytope*,

$$\mathcal{T} \triangleq \{\tau \geq 0 \mid \forall (i,j) \in E, \sum_{x_j} \tau_{ij}(x_i, x_j) = \tau_i(x_i) \text{ and } \forall i \in V, \sum_{x_i} \tau_i(x_i) = 1\}.$$

the reweighted free energy approximation at temperature $T = 1$ is given by

$$\log F_{\text{B},\rho}(G, \tau; \phi, \psi) = U(\tau; \phi, \psi) - H(\tau, \rho)$$

where $U$ is the energy,

$$U(\tau; \phi, \psi) = -\sum_{i \in V} \sum_{x_i} \tau_i(x_i) \log \phi_i(x_i) - \sum_{(i,j) \in E} \sum_{x_i, x_j} \tau_{ij}(x_i, x_j) \log \psi_{ij}(x_i, x_j),$$

and $H$ is an entropy approximation,

$$H(\tau, \rho) = -\sum_{i \in V} \sum_{x_i} \tau_i(x_i) \log \tau_i(x_i) - \sum_{(i,j) \in E} \sum_{x_i, x_j} \rho_{ij} \tau_{ij}(x_i, x_j) \log \frac{\tau_{ij}(x_i, x_j)}{\tau_i(x_i) \tau_j(x_j)}.$$

Here, $\rho_{ij}$ controls the reweighting over the edge $(i, j)$ in the graphical model. If $\rho_{ij} = 1$ for all $(i, j) \in E$, then we call this the Bethe approximation and will typically drop the $\rho$ writing $Z_{\text{B},\bar{1}} = Z_{\text{B}}$. The reweighted partition function is then expressed in terms of the minimum value achieved by this approximation over $\mathcal{T}$ as follows.

$$Z_{\text{B},\rho}(G; \phi, \psi) = e^{-\min_{\tau \in \mathcal{T}} F_{\text{B},\rho}(G, \tau; \phi, \psi)}$$

Similar to the exact partition function computation, the reweighted partition function at temperature $T$ is given by $Z_{\text{B},\rho}(G; \phi^{1/T}, \psi^{1/T})^T$. The zero temperature limit corresponds to minimizing the energy function over the local marginal polytope.

In practice, local optima of these free energy approximations can be found by a reweighted version of belief propagation. The fixed points of this reweighted algorithm correspond to stationary points of $\log Z_{\text{B}}(G, \tau; \phi, \psi)$ over $\mathcal{T}$ [10]. The TRBP algorithm chooses the vector $\rho$ such that $\rho_{ij}$ corresponds to the edge appearance probability of $(i, j)$ over a convex combination of spanning trees. For these choices of $\rho$, the reweighted free energy approximation is convex in $\tau$, $Z_{\text{B},\rho}(G; \phi, \psi)$ is always larger than the true partition function and there exists an ordering of the message updates so that reweighted belief propagation is guaranteed to converge.

## 2.4 Log-Supermodularity and Lower Bounds

A recent theorem of Vontobel [8] provides a combinatorial characterization of the Bethe partition function in terms of graph covers.

**Theorem 2.2** (8)**.**

$$Z_{\mathrm{B}}(G; \phi, \psi) = \limsup_{M \to \infty} \sqrt[M]{\sum_{H \in \mathcal{C}^M(G)} \frac{Z(H; \phi^H, \psi^H)}{|\mathcal{C}^M(G)|}}$$

*where $\mathcal{C}^M(G)$ is the set of all $M$-covers of $G$.*

This characterization suggests that bounds on the partition functions of individual graph covers can be used to bound the Bethe partition function. This approach has recently been used to prove that the Bethe partition function provides a lower bound on the true partition function in certain nice families of graphical models [8, 11, 12]. One such nice family is the family of so-called log-supermodular (aka attractive) graphical models.

**Definition 2.3.** *A function $f : \{0,1\}^n \to \mathbb{R}_{\geq 0}$ is **log-supermodular** if for all $x, y \in \{0,1\}^n$*

$$f(x)f(y) \leq f(x \wedge y)f(x \vee y)$$

*where $(x \wedge y)_i = \min\{x_i, y_i\}$ and $(x \vee y)_i = \max\{x_i, y_i\}$. Similarly, $f$ is **log-submodular** if the inequality is reversed for all $x, y \in \{0,1\}^n$.*

**Theorem 2.4** (Ruozzi [11])**.** *If $(G; \phi, \psi)$ is a log-supermodular graphical model, then for any $M$-cover, $(H; \phi^H, \psi^H)$, of $(G; \phi, \psi)$, $Z(H; \phi^H, \psi^H) \leq Z(G; \phi, \psi)^M$.*

Plugging this result into Theorem 2.2, we can conclude that the Bethe partition function always lower bounds the true partition function in log-supermodular models.

## 3 Switching Log-Supermodular Functions

Let $(G; \phi, \psi)$ be a pairwise binary graphical model. Each $\psi_{ij}$, in this model, is either log-supermodular, log-submodular, or both. In the case that each $\psi_{ij}$ is log-supermodular, Theorem 2.4 says that the partition function of the disconnected 2-cover of $G$ provides an upper bound on the partition function of any other 2-cover of $G$.

When the $\psi_{ij}$ are not all log-supermodular, this is not necessarily the case. As an example, if $G$ is a 3-cycle, then, up to isomorphism, $G$ has two distinct covers: the 6-cycle and the graph consisting of two disconnected 3-cycles. Consider the pairwise binary graphical model for the independent set problem on $G = (V, E)$ given by the edge potentials $\psi_{ij}(x_i, x_j) = 1 - x_i x_j$ for all $(i, j) \in E$. We can easily check that the 6-cycle has 18 distinct independent sets while the disconnected cover has only 16 independent sets. That is, the disconnected 2-cover does not provide an upper bound on the number of independent sets in all 2-covers.

Sometimes graphical models that are not log-supermodular can be converted into log-supermodular models by performing a simple change of variables (e.g., for a fixed $I \subseteq V$, a change of variables that sends $x_i \mapsto 1 - x_i$ for each $i \in I$ and $x_i \mapsto x_i$ for each $i \in V \setminus I$). As a change of variables does not change the partition function, we can directly apply Theorem 2.4 to the new model. We will call such functions switching log-supermodular. These functions are the log-supermodular analog of the "switching supermodular" and "permuted submodular" functions considered by Crama and Hammer [13] and Schlesinger [14] respectively.

The existence of a 2-cover whose partition function is larger than the disconnected one is not unique to the problem of counting independent sets. Such a cover exists whenever the base graphical model is not switching log-supermodular. In this section, we will describe one possible construction of a specific 2-cover that is distinct from the disconnected 2-cover whenever the given graphical model is not switching log-supermodular and will always provide an upper bound on the true partition function.

### 3.1 Signed Graphs

In order to understand when a graphical model can be converted into a log-supermodular model by switching some of the variables, we introduce the notion of a signed graph. A signed graph is

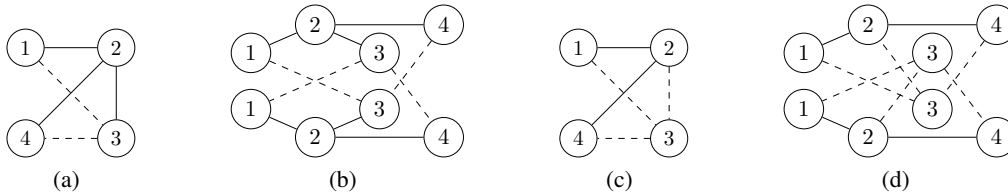

Figure 2: An example of the construction of the 2-cover $G^2$ for the same graph with different edge potentials. Here, dashed lines represent edges with log-submodular potentials. The graph in (b) is the 2-cover construction of the graph in (a) and the graph in (d) is the 2-cover construction applied to the graph in (c). Note that the graph in (b) is connected while the graph in (d) is not.

a graph in which each edge has an associated sign. For our graphical models, we will use a "+" to represent a log-supermodular edge and a "−" to represent a log-submodular edge. The sign of a cycle in the graph is positive if it has an even number of "−" edges and negative otherwise. A signed graph is said to be balanced if there are no negative cycles. Equivalently, a signed graph is balanced, if we can divide its vetices into two sets $A$ and $B$ such that all edges in the graph with one endpoint in set $A$ and the other endpoint in the set $B$ are negative and the remaining edges are positive [15]. Switching, or flipping, a variable as above has the effect of flipping the sign of all edges adjacent to the corresponding variable node in the graphical model: flipping a single variable converts an incident log-supermodular edge into a log-submodular edge and vice versa. A graphical model is switching log-supermodular if and only if its signed graph is balanced.

Signed graphs have been studied before in the context of graphical models. Watanabe [16] characterized signed graphs for which belief propagation is guaranteed to have a unique fixed point. These results depend only on the graph structure and the signs on the edges and not on the strength of the potentials.

## 3.2 Switching Log-Supermodular 2-covers

We can always construct a 2-cover of a pairwise binary graphical model that is switching log-supermodular.

**Definition 3.1.** *Given a pairwise binary graphical model* $(G; \phi, \psi)$, *construct a 2-cover,* $(G^2; \phi^{G^2}, \psi^{G^2})$ *where* $G^2 = (V^{G^2}, E^{G^2})$, *as follows.*

- *For each* $i \in G$, *create two copies of* $i$, *denoted* $i_1$ *and* $i_2$, *in* $V^{G^2}$.

- *For each edge* $(i, j) \in E$, *if* $\psi_{ij}$ *is log-supermodular, then add the edges* $(i_1, j_1)$ *and* $(i_2, j_2)$ *to* $E^{G^2}$. *Otherwise, add the edges* $(i_1, j_2)$ *and* $(i_2, j_1)$ *to* $E^{G^2}$.

$G^2$ is switching log-supermodular. This follows from the characterization of Harary [15] as $G^2$ can be divided into two sets $V_1$ and $V_2$ with only negative edges between the two partitions and positive edges elsewhere. See Figure 2 for an example of this construction.

If all of the potentials in $(G; \phi, \psi)$ are log-supermodular, then $G^2$ is equal to the disconnected 2-cover of $G$. If all of the potentials in $(G; \phi, \psi)$ are log-submodular, then $G^2$ is a bipartite graph.

**Lemma 3.2.** *For a connected graph* $G$, $(G^2; \phi^{G^2}, \psi^{G^2})$ *is disconnected if and only if* $f^{(G; \phi, \psi)}$ *is switching log-supermodular. Equivalently,* $G^2$ *is disconnected if and only if the signed version of* $G$ *is balanced.*

Returning to the example of counting independent sets on a 3-cycle at the beginning of this section, we can check that $G^2$ for this graphical model corresponds to the 6-cycle. The observation that the 6-cycle has more independent sets than two disconnected copies of the 3-cycle is a special case of a general theorem.

**Theorem 3.3.** *For any pairwise binary graphical model* $(G; \phi, \psi)$, $Z(G^2; \phi^{G^2}, \psi^{G^2}) \geq Z(G; \phi, \psi)^2$.

The proof of Theorem 3.3 can be found in Appendix A of the supplementary material. Unlike Theorem 2.4 that provides lower bounds on the partition function, Theorem 3.3 provides an upper bound on the partition function.

# 4 Properties of the Cover $G^2$

In this section, we study the implications that Theorem 2.4 and Theorem 3.3 have for characterizations of switching log-supermodular functions and the computation of the Bethe partition function.

## 4.1 Field Independence

We begin with the simple observation that Theorem 3.3, like Theorem 2.4, does not depend on the choice of external field. In fact, in the case that all of the edge potentials are strictly larger than zero, this independence of external field completely characterizes switching log-supermodular graphical models.

**Theorem 4.1.** *For a pairwise binary graphical model $(G; \phi, \psi)$ with strictly positive edge potentials $\psi$, the following are equivalent.*

1. *$f^{(G; \phi, \psi)}(x)$ is switching log-supermodular.*

2. *For all $M \geq 1$, any external field $\widehat{\phi}$, and any $M$-cover $(H; \widehat{\phi}^H, \psi^H)$ of $(G; \widehat{\phi}, \psi)$, $Z(H; \widehat{\phi}^H, \psi^H) \leq Z(G; \widehat{\phi}, \psi)^M$.*

3. *For all choices of the external field $\widehat{\phi}$ and any 2-cover $(H; \widehat{\phi}^H, \psi^H)$ of $(G; \widehat{\phi}, \psi)$, $Z(H; \widehat{\phi}^H, \psi^H) \leq Z(G; \widehat{\phi}, \psi)^2$.*

In other words, if all of the edge potentials are strictly positive, and the graphical model has even one negative cycle, then there exists an external field $\widehat{\phi}$ and a 2-cover $(H; \widehat{\phi}^H, \psi^H)$ of $(G; \widehat{\phi}, \psi)$ such that $Z(G; \widehat{\phi}, \psi)^2 < Z(H; \widehat{\phi}^H, \psi^H)$. In particular, the proof of the theorem shows that there exists an external field $\widehat{\phi}$ such that $Z(G; \widehat{\phi}, \psi)^2 < Z(G^2; \widehat{\phi}^{G^2}, \psi^{G^2})$. See Appendix B in the supplementary material for a proof of Theorem 4.1.

## 4.2 Bethe Partition Function of Graph Covers

Although the true partition function of an arbitrary graph cover could overestimate or underestimate the true partition function of the base graphical model, the Bethe partition function on every cover always provides an upper bound on the Bethe partition function of the base graph. In addition, the reweighted free energy is always convex for an appropriate choice of parameters $\rho_{TRBP}$ which means that $Z_{\mathrm{B}, \rho_{TRBP}}(G; \phi, \psi)^2 = Z_{\mathrm{B}, \rho_{TRBP}}(G^2; \phi^{G^2}, \psi^{G^2})$. Consequently,

$$Z_{\mathrm{B}, \rho_{TRBP}}(G; \phi, \psi)^2 \geq Z(G^2; \phi^{G^2}, \psi^{G^2}) \geq Z_{\mathrm{B}}(G^2; \phi^{G^2}, \psi^{G^2}) \geq Z_{\mathrm{B}}(G; \phi, \psi)^2. \tag{1}$$

Because the graph cover $G^2$ is switching log-supermodular, the convergence properties of BP are better [5], and we can always apply the PTAS of Weller and Jebara [3] to $(G^2; \phi^{G^2}, \psi^{G^2})$ in order to obtain an upper bound on the Bethe partition function of the original model. That is, by forming the special graph cover $G^2$, we accomplished our stated goal of deriving an algorithm that produces better estimates of the partition function than TRBP but has better convergence properties than BP. We examine the convergence properties experimentally in Section 5.

Before we evaluate the empirical properties of this strategy, observe that (1) holds for the MAP inference problem as well. In the zero temperature limit, computing the Bethe partition function is equivalent to minimizing the energy over the local marginal polytope. Many provably convergent message-passing algorithms have been designed for this specific task [17, 18, 19, 1].

By Theorem 3.3, the MAP solution on $(G^2; \phi^{G^2}, \psi^{G^2})$ is always at least as good as the MAP solution on the original graph. The problem of finding the MAP solution for a log-supermodular pairwise binary graphical model is exactly solvable in strongly polynomial time using max-flow

[20, 21]. We can show that the optimal solution to the Bethe approximation in the zero temperature limit is attained as an integral assignment on this specific 2-cover. The argument goes as follows. The graphical model $(G^2; \phi^{G^2}, \psi^{G^2})$ is switching log-supermodular. By Theorem 2.4, in the zero temperature limit, no MAP solution on any cover of $(G^2; \phi^{G^2}, \psi^{G^2})$ can attain a higher value of the objective function. This means that

$$\lim_{T \to 0} Z_{\mathrm{B}}(G^2; (\phi^{G^2})^{1/T}, (\psi^{G^2})^{1/T})^T = \max_{x_{G^2}} f^{(G^2; \phi^{G^2}, \psi^{G^2})}(x_{G^2}).$$

By (1), the Bethe approximation on $(G^2; \phi^{G^2}, \psi^{G^2})$ is at least as good as the Bethe approximation on the original problem. In fact, they are equivalent in the zero temperature limit: the only part of the Bethe approximation that is not necessarily convex in $\tau$ is the entropy approximation, which becomes negligible as $T \to 0$.

As a consequence, we can compute the optimum of the Bethe free energy in the zero temperature limit in polynomial time without relying on convergent message-passing algorithms. This is particularly interesting as the local marginal polytope for pairwise binary graphical models has an integer persistence property. Given any fractional optimum $\tau$ of the energy, $U$, over the local marginal polytope, if $\tau_i(0) > \tau_i(1)$, then there exists an integer optimum $\tau'$ in the marginal polytope such that $\tau'(0) > \tau'(1)$ [22]. A similar result holds when the strict inequality is reversed. Therefore, we can compute both the Bethe optimum and partial solutions to the exact MAP inference problem simply by solving a max-flow problem over $(G^2; \phi^{G^2}, \psi^{G^2})$.

In this restricted setting, the two cover $G^2$ is essentially the same as the graph construction produced as part of the quadratic pseudo-boolean optimization (QPBO) algorithm in the computer vision community [23]. In this sense, we can view the technique presented in this work as a generalization of QPBO to approximate the partition function of pairwise binary graphical models.

## 5 Experimental Results

In this section, we present several experimental results for the above procedure. For the experiments, we used a standard implementation of reweighted, asynchronous message passing starting from a random initialization and a damping factor of .9. We test the performance of these algorithms on Ising models with a randomly selected external field and various interaction strengths on the edges. We do *not* use the convergent version of TRBP as the message update order is graph dependent and not as easily parallelizable as the reweighted message-passing algorithm [1]. In addition, alternative message-passing schemes that guarantee convergence tend to converge slower than damped reweighted message passing [24]. In some cases where the TRBP parameter choices do not converge, additional damping does help but does not allow convergence within the specified number of iterations.

The first experiment was conducted on a complete cycle on four nodes. The convergence properties of BP have been studied both theoretically and empirically by Mooij and Kappen [5]. As expected, TRBP provides a looser bound on the partition function than BP on the 2-cover and both typically perform worse in terms of estimation than BP on the original graph (when BP converges there). The experimental results are described in Figure 3. In all cases, the algorithms were run until the messages in consecutive time steps differed by less than $10^{-8}$ or until more than $20,000$ iterations were performed (a single iteration consists of updating all of the messages in the model). In general, BP on the 2-cover construction converges more quickly than both BP and TRBP on the original graph. BP failed to converge as the interaction strength decreased past $-.9$. The number of iterations required for convergence of BP on the 2-cover has a spike at the first interaction strength such that $Z_{\mathrm{B}}(G) \neq \sqrt{Z_{\mathrm{B}}(G^2)}$. Empirically, this occurs because of the appearance of new BP fixed points on the two cover that are close to the BP fixed point on the original graph. As the interaction strength increases past this point, the new fixed points further separate from the old fixed points and the algorithm converges significantly faster.

Our second set of experiments evaluates the practical performance of these three message-passing schemes for Ising models on frustrated grid graphs (which arise in computer vision problems), subnetworks of the Epinions social network (the specific subnetworks tested can be found in Appendix D of the supplementary material), and simple four layer graphical models with five nodes per layer

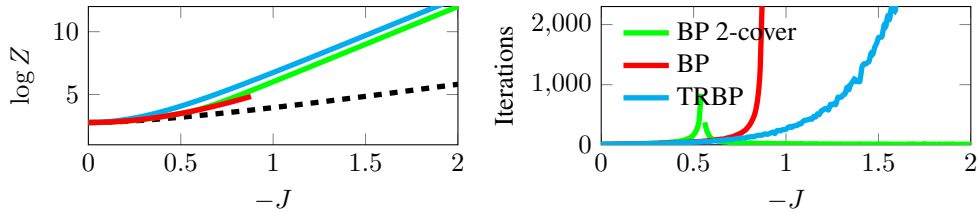

Figure 3: Plots of the log partition function and the number of iterations for the different algorithms to converge for a complete graph on four nodes with no external field as the strength of the negative edges goes from 0 to -2. For TRBP, $\rho_{ij} = .5$ for all $(i,j) \in E$. The dashed black line is the ground truth.

| | $a$ | BP | TRBP | BP 2-cover | BP Iter. | TRBP Iter | BP 2-cover Iter. |
|---|---|---|---|---|---|---|---|
| | 1 | **100%** | **100%** | 95% | **44.62** | 110.41 | 222.99 |
| Grid | 2 | 15% | 30% | **100%** | 210 | 815.3 | **44.14** |
| | 4 | 1% | 0% | **100%** | 219 | - | **29.59** |
| | 1 | 47% | 0% | **100%** | 63.53 | - | **21.12** |
| EPIN1 | 2 | 37% | 0% | **100%** | 90.1 | - | **16.19** |
| | 4 | 38% | 0% | **100%** | 93.63 | - | **15.9** |
| | 1 | 41% | 0% | **100%** | 51.8 | - | **15.12** |
| EPIN1 | 2 | 50% | 0% | 99% | 42.46 | - | **14.84** |
| | 4 | 53% | 0% | **100%** | 86.66 | - | **14.93** |
| | 1 | 61% | 0% | **100%** | 89.2 | - | **16.67** |
| Deep Networks | 2 | 61% | 0% | **100%** | 30.66 | - | **16.82** |
| | 4 | 60% | 0% | **100%** | 24.88 | - | **18.17** |

Figure 4: Percent of samples on which each algorithm converged within 1000 iterations and the average number of iterations for convergence for 100 samples of edges weights in $[-a, a]$ for the designated graphs. For TRBP, performance was poor independent of the spanning trees selected.

similar to those used to model "deep" belief networks (layer $i$ and layer $i + 1$ form a complete bipartite graph and there are no intralayer edges). In the Epinions network, the pairwise interactions correspond to trust relationships. If our goal was to find the most trusted users in the network, then we could, for example, compute the marginal probability that each user is trusted and then rank the users by these probabilities. For each of these models, the edge weights are drawn uniformly at random from the interval $[-a, a]$. The performance of BP, TRBP, and BP on the 2-cover continue to behave as they did for the simple four node model: as $a$ increases, BP fails to converge and BP on the 2-cover converges much faster and more frequently than the other methods. Here, convergence was required to an accuracy of $10^{-8}$ within $1,000$ iterations. The results for the different graphs appear in Figure 4. Notably, both BP and TRBP perform poorly on the real networks from the Epinions data set.

## Acknowledgments

This work was supported in part by NSF grants IIS-1117631, CCF-1302269 and IIS-1451500.

## Footnotes

[1]In the Epinions network, users are connected by agreement and disagreement edges and therefore frustrated cycles abound. By treating the network as a pairwise binary graphical model, we may compute the trustworthiness of a user by performing marginal inference over a variable representing if the user is trusted or not.

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
