[Supplementary Material]

# Supplemental Appendix

For the proofs in this appendix, it will be convenient to express $f^{(H;\phi^H,\psi^H)}$ as a function over $M$ vectors in the set $\{0,1\}^{|V|}$. We can partition the vertex set $V^H$ into $M$ disjoint sets $V_1, \ldots, V_M$ such that each set contains exactly one copy of each vertex in the graph $G$. Then, without loss of generality, any $x \in \{0,1\}^{M|V|}$ can be expressed as $x^1, \ldots, x^m \in \{0,1\}^{|V|}$ where $x^m$ is an assignment to the variables in $V_m$ for all $m \in \{1, \ldots, M\}$. In this case, we will write $f^{(H;\phi^H,\psi^H)}(x) = f^{(H;\phi^H,\psi^H)}(x^1, \ldots, x^M)$.

## A    Proof of Theorem 3.3

Before proving Theorem 3.3, we prove a useful inequality for log-supermodular functions from which the desired theorem will easily follow. This approach is similar to that used by Ruozzi [11] in the proof of Theorem 2.4 but is much simpler as we only need 2-cover inequalities.

**Theorem A.1.** *Let $f_1, f_2 : \{0,1\}^n \to \mathbb{R}_{\geq 0}$ and $g : \{0,1\}^{2n} \to \mathbb{R}_{\geq 0}$ such that $g$ is log-supermodular. If for all $x, y \in \{0,1\}^n$*

$$f_1(x)f_2(y) \leq g(x \vee y, \overline{x \wedge y}),$$

*then*

$$\left[\sum_x f_1(x)\right]\left[\sum_y f_2(y)\right] \leq \sum_{x,y} g(x,y).$$

*Proof.* The proof of the theorem follows by induction on $n$. We begin by showing that the result holds for the case $n = 1$. By assumption, we have the following inequalities.

$$f_1(1)f_2(1) \leq g(1,0)$$
$$f_1(0)f_2(1) \leq g(1,1)$$
$$f_1(1)f_2(0) \leq g(1,1)$$
$$f_1(0)f_2(0) \leq g(0,1)$$

Similarly, the following inequality follows from the above inequalities and the log-supermodularity of $g$.

$$[f_1(1)f_2(0)][f_1(0)f_2(1)] = [f_1(1)f_2(1)][f_1(0)f_2(0)]$$
$$\leq g(1,0)g(0,1)$$
$$\leq g(1,1)g(0,0)$$

Combining these two sets of inequalities and using the observation that weak log-majorization implies weak majorization yields the desired result for the base case [25].

The remainder of the proof now follows by induction on $n$. Let $n \geq 2$ and suppose that the result holds for all $n - 1$. Let $f_1, f_2 : \{0,1\}^n \to \mathbb{R}_{\geq 0}$ and $g : \{0,1\}^{2n} \to \mathbb{R}_{\geq 0}$ be nonnegative real-valued functions such that $g$ is log-supermodular. Further, suppose that these functions satisfy the assumptions of the theorem.

Define $f' : \{0,1\}^{n-1} \to \mathbb{R}_{\geq 0}$ and $g' : \{0,1\}^{2(n-1)} \to \mathbb{R}_{\geq 0}$ as

$$f_i'(y) = f_i(y,0) + f_i(y,1)$$
$$g'(y^1, y^2) = \sum_{s^1,s^2 \in \{0,1\}} g(y^1, s^1, y^2, s^2)$$

Notice that $g'$ is log-supermodular because it is the marginal of a log-supermodular function. If we can show that

$$f_1'(y^1)f_2'(y^2) \leq g'(y^1 \vee y^2, \overline{y^1 \wedge y^2})$$

for all $y^1, y^2 \in \{0,1\}^{n-1}$, then the result will follow by induction on $n$.

To show this, fix $z^1, z^2 \in \{0,1\}^{n-1}$ and define $f'' : \{0,1\} \to \mathbb{R}_{\geq 0}$ and $g'' : \{0,1\}^2 \to \mathbb{R}_{\geq 0}$ as

$$f_i''(s) = f_i(z^i, s)$$
$$g''(s^1, s^2) = g(z^1 \vee z^2, s^1, \overline{z^1 \wedge z^2}, s^2).$$

We can easily check that $g''(s^1, s^2)$ is log-supermodular and that $g''(s^1 \vee s^2, \overline{s^1 \wedge s^2}) \geq f_1''(s^1) f_1''(s^2)$ for all $s^1, s^2 \in \{0,1\}$. Hence, by the base case,

$$
\begin{aligned}
f_1'(z^1) f_2'(z^2) &= \sum_{s^1, s^2 \in \{0,1\}} f_1''(s^1) f_2''(s^2) \\
&\leq \sum_{s^1, s^2 \in \{0,1\}} g''(s^1, s^2) \\
&= g'(z^1 \vee z^2, \overline{z^1 \wedge z^2})
\end{aligned}
$$

which completes the proof of the theorem.

$\square$

We now use Theorem A.1 in order to prove Theorem 3.3.

**Theorem.** *For any pairwise binary graphical model* $(G; \phi, \psi)$, $Z(G^2; \phi^{G^2}, \psi^{G^2}) \geq Z(G; \phi, \psi)^2$.

*Proof.* For each $(i,j) \in E$, either $\psi_{ij}$ is log-supermodular, in which case

$$\psi_{ij}(x_i^1, x_j^1)\psi_{ij}(x_i^2, x_j^2) \leq \psi_{ij}(x_i^1 \vee x_i^2, x_j^1 \vee x_j^2)\psi_{ij}(x_i^1 \wedge x_i^2, x_j^1 \wedge x_j^2)$$

for all $x^1, x^2 \in \{0,1\}$, or log-submodular, in which case

$$\psi_{ij}(x_i^1, x_j^1)\psi_{ij}(x_i^2, x_j^2) \leq \psi_{ij}(x_i^1 \vee x_i^2, x_j^1 \wedge x_j^2)\psi_{ij}(x_i^1 \wedge x_i^2, x_j^1 \vee x_j^2)$$

for all $x^1, x^2 \in \{0,1\}$. Applying these inequalities to the disconnected 2-cover yields

$$f^{(G;\phi,\psi)}(x^1) f^{(G;\phi,\psi)}(x^2) \leq f^{(G^2;\phi,\psi)}(x^1 \vee x^2, x^1 \wedge x^2).$$

Now, define the log-supermodular switching of $f^{(G^2;\phi,\psi)}$ as $g(x^1, x^2) \triangleq f^{(G^2;\phi,\psi)}(x^1, \overline{x^2})$ for all $x^1, x^2 \in \{0,1\}^n$. This gives

$$f^{(G;\phi,\psi)}(x^1) f^{(G;\phi,\psi)}(x^2) \leq g(x^1 \vee x^2, \overline{x^1 \wedge x^2}) \tag{2}$$

for all $x^1, x^2 \in \{0,1\}^n$. Applying Theorem A.1 to (2), yields $Z(G; \phi, \psi)^2 \leq Z(G^2; \phi^{G^2}, \psi^{G^2})$ as desired. $\square$

# B  Proof of Theorem 4.1

*Proof.* $(1 \Rightarrow 2)$ follows from by switching $f^{(G;\phi,\psi)}$ to a log-supermodular function and then invoking Theorem 2.4, and $(2 \Rightarrow 3)$ is trivial.

For the remaining implication $(3 \Rightarrow 1)$ suppose by way of contradiction, that $f^{(G;\phi,\psi)}$ is not switching log-supermodular (i.e., the graphical model contains a negative cycle). We can assume that the model contains no edge potentials that are simultaneously log-supermodular and log-submodular as these potentials can always be written as a product of two self-potentials over different variables and absorbed into other edge potentials without affecting the signs on the edges of the graph [26].

Pick one negative cycle in the model. Denote it by $C = (V_C, E_C)$. We can assume that the cycle $C$ is cordless. If not, we can find a smaller negative cycle that uses one of the cords.

As the external field does not affect the existence of negative cycles, we are free to choose it as we please. Construct an external field $\widehat{\phi}$ as follows. For each $k \in V \setminus V_C$, $\widehat{\phi}_k(1) = 1$ and $\widehat{\phi}_k(0) = 0$. For each $i \in V_C$ and $x_i \in \{0,1\}$,

$$\widehat{\phi}_i(x_i) = \frac{1}{\prod_{k \in \partial i \setminus V_C} \widehat{\psi}_{ik}(x_i, 1)}.$$

This external field effectively reduces the problem of computing the partition function of the entire graph to that of computing the partition function over only the cycle $V_C$ in the absence of an external field.

$$Z(G; \widehat{\phi}, \psi) = \kappa Z(C; \phi^{\text{const}}, \psi_{E_C})$$

Here, $\phi^{\text{const}}$ is a constant external field, $\kappa$ is a positive constant, and $\psi_{E_C}$ is the restriction of $\psi$ to the cycle $C$. A similar expression holds for the graphical model $(G^2; \widehat{\phi}^{G^2}, \psi^{G^2})$.

$$Z(G^2; \widehat{\phi}^{G^2}, \psi^{G^2}) = \kappa^2 Z(C^2; \phi^{\text{const}}, \psi_{E_C}^{G^2})$$

To complete the proof, we will show that

$$Z(C; \phi^{\text{const}}, \psi_{E_C})^2 < Z(C^2; \phi^{\text{const}}, \psi_{E_C}^{G^2})$$

in contradiction of bullet 3 of the theorem. To see this, we need a few observations about the computation of the partition function on a cycle. First, we will represent each potential function $\psi_{ij}$ as a matrix $A^{ij} \in \mathbb{R}_{\geq 0}^{2 \times 2}$ where $A_{ab}^{ij} = \psi_{ij}(a-1, b-1)$ for all $a, b \in \{1, 2\}$.

By assumption $C$ does not contain any edge potentials that are both log-submodular and log-supermodular, so we must have that, for each $(i, j) \in E_C$, the sign of the edge corresponds to the determinant of $A^{ij}$. Further note that, by the Perron-Frobenius Theorem and the fact that $A^{ij}$ is a nonnegative matrix, $A^{ij}$ must have either two positive eigenvalues or one positive eigenvalue and one negative eigenvalue. From this, we can conclude that every $A^{ij}$ corresponding to a negative edge has one positive and one negative eigenvalue while every $A^{ij}$ corresponding to a positive edge must have two positive eigenvalues.

Now, pick some $i \in V_C$ and walk in one direction around the cycle labeling the vertices, starting with $i$, successively as $c_1, ..., c_{|V_C|}$. With these definitions, we have

$$Z(C; \phi^{\text{const}}, \psi_{E_C}) = \sum_{x_C} \prod_{(i,j) \in E_C} \psi_{ij}(x_i, x_j)$$
$$= \sum_{x_C} \prod_{(i,j) \in E_C} A_{x_i, x_j}^{ij}$$
$$= \text{trace}(A^{c_1 c_2} ... A^{c_{|V_C|} c_1}).$$

Denote the matrix product $A^{c_1 c_2} ... A^{c_{|V_C|} c_1}$ simply as the matrix $A_C$. Note that the sign of $\det(A_C)$ is equal to the sign of the cycle (which is negative by assumption). Similarly, for the two cover $C^2$,

$$Z(C^2; \phi^{\text{const}}, \psi_{E_C}^{G^2}) = \text{trace}(A_C A_C).$$

Finally, denote the eigenvalues of $A_C$ as $\lambda_1$ and $\lambda_2$. From the arguments above, exactly one of these is positive and one is negative. This completes the proof as

$$\text{trace}(A_C A_C) = \lambda_1^2 + \lambda_2^2 > (\lambda_1 + \lambda_2)^2 = \text{trace}(A_C)^2.$$

$\square$

## C   Marginal Probability Experiment

In this experiment, we analyze the performance of BP, TRBP, and BP on the 2-cover for marginal estimation. Figure 5 shows the error in the singleton marginals for the experiment in Figure 3 with a randomly chosen external field.

## D   Epinions Social Network

For the Epinions experiments, we randomly generated node induced subgraphs from the Epinions network data collected by Richardson et al. [27]. Figure 6 shows two graphs generated in this way.

Figure 5: Plots of the error between the true marginals and the approximate marginals for a complete graph on four nodes with no external field as the strength of the negative edges goes from 0 to -2. For TRBP, $\rho_{ij} = .5$ for all $(i,j) \in E$. The error is computed as the 2-norm between the vector of true singleton marginals evaluated at one and the approximate singleton marginals evaluated at one.

(a) EPIN1: 92 nodes, 482 edges

(b) EPIN2: 80 nodes, 185 edges

Figure 6: Node-induced subgraphs of the Epinions network.