[Reviews · NeurIPS 2014]

Submitted by Assigned_Reviewer_23

Making Pairwise Binary Graphical Models Attractive

Quality - there are some surprising aspects to the experiments that are not discussed. See below.
Clarity - paper covers a lot of ground many of the subparts are relatively well written, with references where necessary. But I found it surprisingly hard to get quickly a succinct overview of what is achieved. What would help is to emphasize in abstract and intro which claimed improvements are wrt approximating the Bethe free energy (in the direction of recent work by Weller and Jebara) and which wrt the true free energy. E.g. just from the introduction we get the statement "We then show that the computation of the Bethe partition function can approximated or in some cases found exactly by computing the Bethe partition function over this special cover." and " In these experiments, the proposed scheme converges significantly more frequently than reweighted BP and provides a better estimate of the partition function than TRBP." The first is about approximations of the Bethe free energy, the second about the true free energy. Similarly for the assumptions about the model. Is the assumption of switching log-submodular models / signed graphs necessary?
Originality - the paper follows a recent direction, but appears to make clear new contributions.
Significance - finding better approximate inference algorithms is one of the most important challenges in the field. This paper makes advances and increases our understanding in a relatively fresh direction.

Questions:
- In Fig 5 in the appendix the BP-2 curve is non-smooth (assuming it is occluded by the other curve in the left part). Why is this?
- Fig 3 seems to show a weird asymptote for BP-2 this does not seem to be discussed in the text. Did I miss this? What is going on here?
Summary: new insights and extensions to a recent direction to approximate inference. The presentation is not as easy to follow as one might hope. Presentation should be significantly improved to highlight essential assumptions and possibly restricted applicability in abstract and introduction and possibly title.

Submitted by Assigned_Reviewer_43

The authors proposed a new method to compute an upper bound of the partition function of a pairwise binary graphical model. The proposed algorithm has a better convergence property than BP applied to the original graph, and the bound is better than the TRP. In the method, a 2-cover attractive graphical model is generated from the original model and BP is applied since the convergence property of BP is better for attractive graphical models. The authors studied the Bethe free energy of the M-cover model and proved its relation to the true partition function.

The proposed method is new and interesting. Its comparison to the TRP is not so important but there might be a lot of extensions related to this method.

Two questions:
* Why only 2-cover? It seems M-cover can be used, too.
* Do you have any results related to statistical physics?
Summary: The proposed method is new and interesting. Its comparison to the TRP is not so important but there might be a lot of extensions related to this method.

Submitted by Assigned_Reviewer_44

The paper describes a graph construction based on pairwise graph models (2-cover switching log-supermodular). Running BP on that graph has better convergence than on the original, and the obtained results have better estimates of the partition function than TRBP.

Quality - the paper is mostly theoretical, and could use a bit of polishing to crystallize its contribution (for instance what is the practical implication of theorem 4.1, and the conclusion section is missing), but overall it is interesting contribution towards better understanding of message passing algorithms and how to arrive at both empirically good algorithms with good convergence guarantees.

Clarity - the paper is clearly written, there are still a few typos like "deriving and algorithm".

Originality - the paper draws on several existing theoretical findings, using it to draw a contribution which is novel.

Significance - it is unclear how practical this is for and it would be nice to have a discussion of how scalable the approach is. The amount of data used for Epinions experiments is less than 100 points.

Changing the graph colors so that it isn't using two shades of blue is recommended.
What is the reason for the behavior of the BP on 2-cover around -J=0.5?

Summary: A new BP variant with better convergence than regular BP and better log partition function approzimation than provably convergent variants such as TRBP. Based on a graph construction derived from the original pairwise potential graph, doubling the variables and connecting pairs based on their log-supermodularity properties.
Author Feedback
Author rebuttal: We thank the reviewers for their helpful comments and questions. We will implement all their valuable suggestions for improving the presentation and the readability of the text and the figures. For clarity, the practical contributions of this work can be summarized as follows.

We present a general scheme for approximate inference in pairwise binary graphical models. The scheme requires constructing an “attractive” 2-cover of the base graph and performing belief propagation on this cover. The extra overhead in both time in space for this algorithm is only twice that of standard belief propagation, but the algorithm exhibits much better convergence properties on the chosen 2-cover.

Below we address the primary questions and comments of the reviewers.

“Is the assumption of switching log-submodular models / signed graphs necessary?”

The base graphical model can be an arbitrary binary pairwise model. The 2-cover we output from the base model is chosen to be switching log-supermodular since BP is known to perform better on these models.

“Fig 3 seems to show a weird asymptote for BP-2 this does not seem to be discussed in the text. Did I miss this? What is going on here?” & “What is the reason for the behavior of the BP on 2-cover around "-J=0.5”

This asymptote appears to occur whenever the model goes from switching log-supermodular to one with frustrated cycles. For the four node model in the experiments, we verified that the slowdown occurs because of the appearance of new fixed points of the Bethe approximation on the 2-cover. Initially, these new fixed points are very close to the previous fixed points. This seems to be responsible for the slower rates of convergence near this transition point. See line 369 of the submission.

“In Fig 5 in the appendix the BP-2 curve is non-smooth (assuming it is occluded by the other curve in the left part). Why is this?”

Similar to the previous question, the appearance of new fixed points on the two cover means that the algorithm, at some point, may converge to these new fixed points. When this happens, the curve can change shape. Note that this does not happen for the estimate of Z_B because the new fixed points, initially, have nearly the same value as the old ones. However, the pseudomarginals at the new fixed points do not split away from the old fixed points and this produces the non-smooth results in Figure 5.

“Why only 2-covers? It seems M-covers can be used, too.”

We focused on a specific 2-cover because it produces an attractive graphical model (on which BP has better convergence properties). Every graph cover of this 2-cover is also an attractive graphical model, so the theory would extend to all of these. The convergence
properties of BP may also improve over a random M-cover as well, but understanding the properties of random M-covers is significantly more challenging.

“It is unclear how practical this is for and it would be nice to have a discussion of how scalable the approach is.”

It is worth pointing out that, no matter which M-cover is selected for the algorithm, the message passing on this cover can be simulated over the original graphical model where each node plays the role of its M copies (hence, the approach could easily be implemented in decentralized applications such as sensor networks). The additional complexity of the new message-passing scheme scales as M times that of the original model. For the proposed 2-cover, this doubles the amount of computation and storage per iteration. Hence, if BP was practical for the base model, then it should continue to be practical for the larger model. As an example, a naïve implementation in MATLAB for an Epinons graph with around 1500 nodes only requires a few minutes to
converge.